# Machine learning based prediction of recurrence after curative resection for rectal cancer

**Youngbae Jeon**[1‡], **Young-Jae Kim**[2‡], **Jisoo Jeon**[2], **Kug-Hyun Nam**[1], **Tae-Sik Hwang**[1], **Kwang-Gi Kim**[2]*, **Jeong-Heum Baek**[1]*

1 Department of Surgery, Division of Colon and Rectal Surgery, Gil Medical Center, Gachon University College of Medicine, Incheon, South Korea, 2 Department of Biomedical Engineering, Gachon University, Incheon, South Korea

‡ YJ and YJK equally contributed as first authors.
* kimkg@gachon.ac.kr (KGK); gsbaek@gilhospital.com (JHB)

## Abstract

### Purpose

Patients with rectal cancer without distant metastases are typically treated with radical surgery. Post curative resection, several factors can affect tumor recurrence. This study aimed to analyze factors related to rectal cancer recurrence after curative resection using different machine learning techniques.

### Methods

Consecutive patients who underwent curative surgery for rectal cancer between 2004 and 2018 at Gil Medical Center were included. Patients with stage IV disease, colon cancer, anal cancer, other recurrent cancer, emergency surgery, or hereditary malignancies were excluded from the study. The Synthetic Minority Oversampling Technique with Tomek link (SMOTETomek) technique was used to compensate for data imbalance between recurrent and no-recurrent groups. Four machine learning methods, logistic regression (LR), support vector machine (SVM), random forest (RF), and Extreme gradient boosting (XGBoost), were used to identify significant factors. To overfit and improve the model performance, feature importance was calculated using the permutation importance technique.

### Results

A total of 3320 patients were included in the study. After exclusion, the total sample size of the study was 961 patients. The median follow-up period was 60.8 months (range:1.2–192.4). The recurrence rate during follow-up was 13.2% (n = 127). After applying the SMOTETomek method, the number of patients in both groups, recurrent and non-recurrent group were equalized to 667 patients. After analyzing for 16 variables, the top eight ranked variables {pathologic Tumor stage (pT), sex, concurrent chemoradiotherapy, pathologic Node stage (pN), age, postoperative chemotherapy, pathologic Tumor-Node-Metastasis stage (pTNM), and perineural invasion} were selected based on the order of permutational importance. The highest area under the curve (AUC) was for the SVM method (0.831). The

**Data Availability Statement:** All relevant data are within the paper and its Supporting Information files.

**Funding:** The authors received no specific funding for this work.

**Competing interests:** The authors have declared that no competing interests exist.

sensitivity, specificity, and accuracy were found to be 0.692, 0.814, and 0.798, respectively. The lowest AUC was obtained for the XGBoost method (0.804), with a sensitivity, specificity, and accuracy of 0.308, 0.928, and 0.845, respectively. The variable with highest importance was pT as assessed through SVM, RF, and XGBoost (0.06, 0.12, and 0.13, respectively), whereas pTNM had the highest importance when assessed by LR (0.05).

## Conclusions

In the current study, SVM showed the best AUC, and the most influential factor across all machine learning methods except LR was found to be pT. The rectal cancer patients who have a high pT stage during postoperative follow-up are need to be more close surveillance.

## Introduction

Colorectal cancer is a common malignant disease having the third highest incidence and second highest mortality rates worldwide [1]. Rectal cancer, accounts for approximately one-third of all colorectal cancers and has a relatively higher recurrence rates than colon cancer. This is due to the lower rectum being devoid of serosa which protects against tumor invasion through the muscle layer, and it is also technically more demanding to obtain a sufficient safety margin [2]. The 5-year recurrence rate of locally advanced rectal cancer after curative surgery is reported to be in the range of 6–27.5% [3]. Such a high rate is associated with both tumor- and treatment-related factors. Early detection and immediate treatment of rectal cancer recurrence may prevent patients from entering a dismal stage. Therefore, clinicians need to identify the factors that increase the risk of rectal cancer recurrence and be more alert during the follow-up period after surgery.

In the recent years, artificial intelligence has been in the spotlight in varied fields, with its applications in the medical field rapidly progressing. Machine learning based algorithms, which forms the basis of artificial intelligence, have been developed over the past decades for predicting disease risk, prognosis, diagnosis, and even the course of treatment in healthcare settings [4]. Further, recent studies have reported the feasibility and utility of artificial intelligence-based predicting the recurrence of several malignant diseases, including colorectal, breast, and gastric cancer [5–10]. However, in colorectal cancer, only a few studies employing machine-learning methodologies focus exclusively on recurrence prediction for rectal cancer without including colon cancer. Hence, we aimed to compare four different machine learning algorithms in terms of performance and accuracy in predicting significant risk factors for the recurrence of rectal cancer after curative resection.

## Materials and methods

### Patient selection and dataset

We used the colorectal cancer surgery database, which was retrospectively collected from the Clinical Research Data Warehouse (CRDW) at the Gil Medical Center. The data were accessed for research purpose since August 27, 2021. All data has been anonymized so that individual participant could not be identified. The database included 3320 consecutive patients who underwent surgery for colorectal cancer between January 2004 and December 2018. From the databases, we identified patients who underwent curative surgery (R0 or R1 resection which means tumor free resection margin or resection margin with microscopic residual tumor and without macroscopic residual tumor) for rectal cancer. Patients with stage IV disease, colon

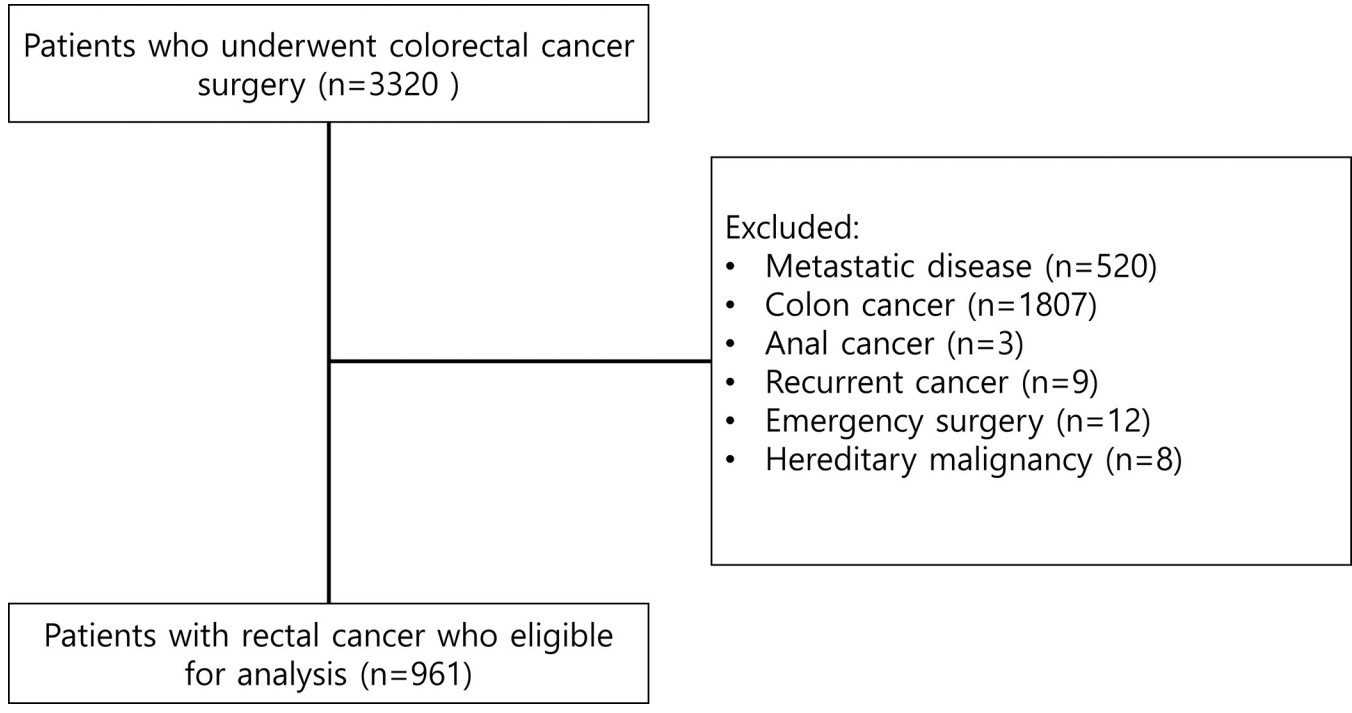

**Fig 1. Flowchart of the patient selection.**

cancer, anal cancer, recurrent cancer, emergency surgery, or hereditary malignancies were excluded from the study. After exclusion, 961 patients remained eligible for the study (Fig 1). All of the included patients underwent total mesorectal excision by open or laparoscopic approach depended on surgeon's preference. The patients who have clinical tumor stage 3–4 (cT3-4) with any clinical node stage (cNany) or any clinical tumor stage (cTany) with clinical positive node stage (cN1-2) received concurrent chemoradiotherapy (CCRT) at 8 to 10 weeks before surgery. Adjuvant chemotherapy was determined by multidisciplinary discussion considering final pathologic stage, and patient's clinical condition. There were 834 and 127 patients in the no-recurrence and recurrence groups, respectively. For model training, the overall database was divided into training and testing datasets. Randomly selected each 20% of data from the recurrence and no-recurrence groups were used as the test dataset (n = 193), and the remaining data were used as a training dataset (n = 768).

## Ethics and consent

This study obtained institutional review board approval from the Ethics Review Committee of the Gil Medical Center (approval no. GAIRB2021-316). All procedures were performed in accordance with the ethical standards of Gil Medical Center at Gachon University, and the 1964 Declaration of Helsinki and its later amendments. Because of the retrospective nature of the study, the need to obtain informed consent was waived for the individual participants by the Ethics Review Committee.

## Compensating for data imbalances

In this study, we employed the Synthetic Minority Oversampling Technique with Tomek link (SMOTETomek) technique to address the data imbalance issue between the recurrence and

no-recurrence groups. SMOTETomek combines oversampling and under sampling techniques, utilizing SMOTE for oversampling and the Tomek link for under sampling. SMOTE employs the k-nearest neighbor (KNN) algorithm to identify minority classes and generates new samples with randomly assigned values ranging from 0 to 1. The Tomek link eliminates samples belonging to the majority class from pairs of neighboring samples of different classes [11]. By utilizing the SMOTETomek technique, we sampled 1334, with 667 in the relapsed group and 667 in the non-relapsed group, effectively addressing and accounting for the data imbalance.

## Potential predictors

The database included 43 clinical features, and surgeons initially selected 16 features that were considered clinically related to rectal cancer recurrence. The following features were analyzed by the machine learning techniques: patient baseline characteristics {age, sex, American Society of Anesthesiologists score (ASA), body mass index (BMI), and initial carcinoembryonic antigen (CEA)}, treatment related factors (CCRT, and postoperative chemotherapy), and tumor related factors {location of rectal cancer, histologic type, pathologic Tumor stage (pT), pathologic Node stage (pN), pathologic Tumor-Node-Metastasis stage (pTNM), lymphovascular invasion (LVI), perineural invasion (PNI), involvement of distal resection margin, and harvested lymph nodes). Tumor stage was defined according to the American Joint Committee on Cancer (AJCC) 8$^{th}$ edition [12]. All continuous variables were converted to incategorical variables according to their clinical significance: Age was divided into $< 65$, and $\geq 65$ years; BMI was divided into $< 25$, and $\geq 25$ kg/m2; Initial CEA was divided into $< 5$, and $\geq 5$ng/ml; The number of harvested lymph nodes was divided into $< 12$, and $\geq 12$. None of the included variables had any missing values.

## Machine learning algorithms

Logistic regression (LR) is an algorithm that applies a logistic function to the coefficients obtained from linear regression to classify the values. It uses a linear combination of each independent variable to make a probability prediction and is classically and widely used to identify risk factors in medical research [13]. Support vector machine (SVM) is an algorithm that converts input data into high-dimensional spatial data and then determines the optimal decision boundary that maximizes the distance between data classes [14]. Further, Random Forest (RF) is an ensemble model that builds on the Decision Tree model. It creates multiple decision trees and aggregates the results of each tree using an ensemble technique to make a final decision [15]. Extreme gradient boosting (XGBoost) is an algorithm that addresses the shortcomings of the Gradient Boosting algorithm and is known for its speed and superior prediction performance compared with other models. Internal cross-validation was performed at each iteration to prevent overfitting [16].

## Feature selection

In this study, we employed a permutation-importance technique for the feature selection. Permutation importance is a method commonly used in machine learning to assess the significance of model features, offering the advantage of applicability to any type of model. This technique quantifies the increase in prediction error when the values of the features are randomly permuted, thus breaking the relationship between the features and the actual outcome. By observing the increase in the model error for each feature, we gained some insights into the dependency of that particular attribute [17]. We utilized permutation importance to select

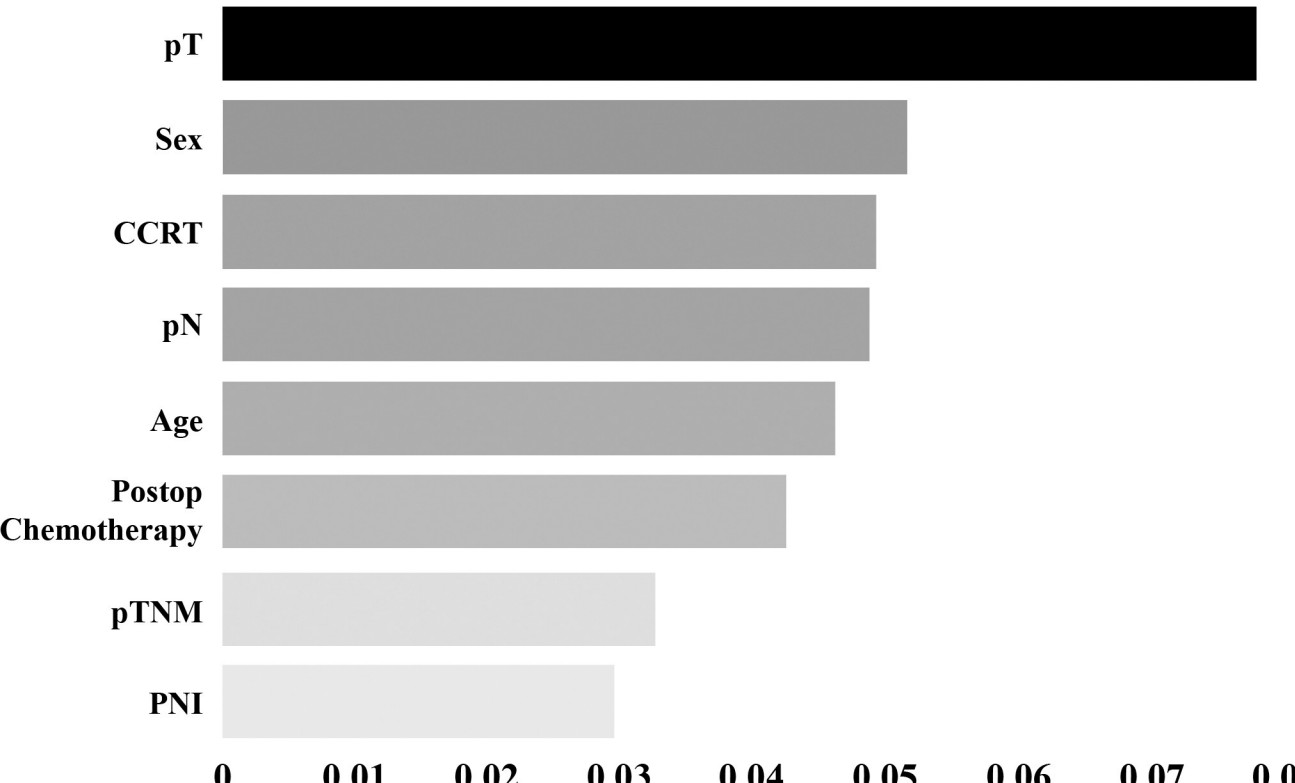

**Fig 2. Eight high-ranked features for rectal cancer recurrence after curative surgery by the mean value of permutation importance in four machine learning methods.**

features from a pool of 16, ultimately identifying 8 key features: PNI, pTNM, postoperative chemotherapy, age, pN, CCRT, sex, and pT (Fig 2).

## Optimal combination of hyperparameter

In this study, we used a grid search technique to tune the hyperparameters of each machine learning model. A grid search is an exploratory technique that determines the optimal combination of hyperparameter values by exploring all possible combinations [18]. We utilized a grid search to combine hyperparameter values for each model and cross-validated each combination using the training data to select the parameter combination exhibiting the best area under the curve (AUC) performance.

## Model performance comparison

After feature selection based on permutation importance, four machine learning algorithms were trained with selected features of the training dataset (n = 1334). For model performance comparison, the following indices were used: sensitivity, specificity, accuracy, and AUC.

For machine learning, statistical analysis, and performance validation, we used Python software (version 3.7.0; Python Software Foundation, Wilmington, DE, USA) and the scikit-learn library (version 0.23.2). Statistical Package for the Social Sciences (SPSS, version 20, IBM Corp., Armonk, NY, USA) was used for the analysis. Statistical significance was set at $p < 0.05$. A schematic flowchart of the study is shown in Fig 3.

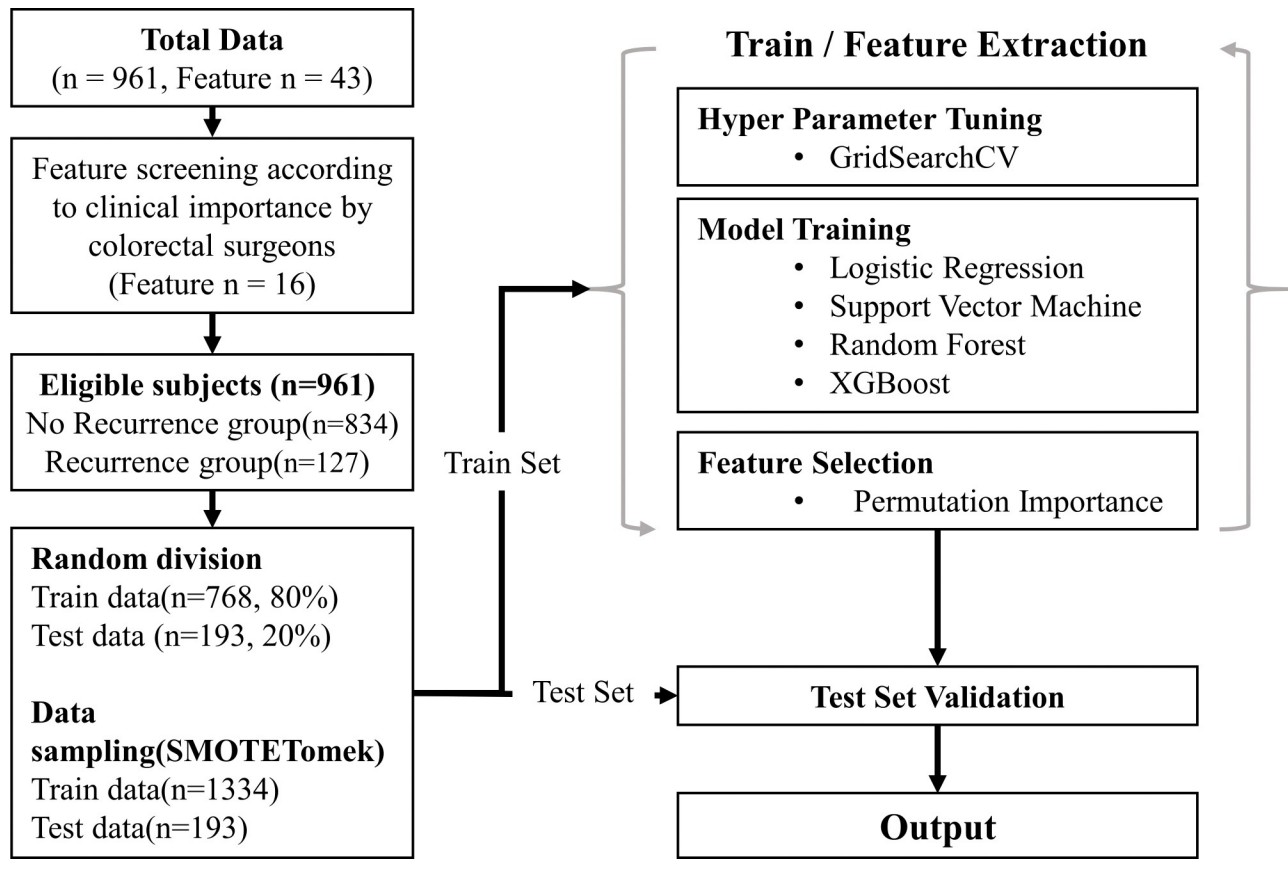

**Fig 3. Schematic flow chart of the study.**

## Results

### Baseline patient demographics

A total of 961 patients were included in the study. The median follow-up period was 60.8 months (range:1.2–192.4). The recurrence rate during follow-up was 13.2% (n = 127). In the chi-square test, age, initial CEA level, pT, LVI, PNI, pN, pTNM, and postoperative chemotherapy were statistically significant ($p < 0.05$). The baseline patient demographics are shown in Table 1.

### Model performance outcomes

The highest AUC was obtained for SVM (0.831, 95% confidence interval:0.770–0.881). The sensitivity, specificity, and accuracy were 0.692 (95% confidence interval:0.482–0.857), 0.814 (95% confidence interval:0.747–0.870), and 0.798 (95% confidence interval:0.734–0.852), respectively. The lowest AUC was observed for XGBoost (0.804; 95% confidence interval:0.741–0.857), and its sensitivity, specificity, and accuracy were 0.308 (95% confidence interval:0.143–0.518), 0.928 (95% confidence interval:0.878–0.962), and 0.845 (95% confidence interval:0.786–0.918), respectively. In terms of the AUC value, SVM showed the best performance, whereas the specificity and accuracy were the highest for XGBoost. The confusion matrix for the model performance comparison and receiver operating characteristic (ROC) curves are shown in Table 2 and Fig 4.

**Table 1. The baseline patient demographics.**

| Feature | No Recurrence (n = 834) | Recurrence (n = 127) | P value |
|---|---|---|---|
| Age | | | 0.0071 |
| <65 | 428 (51.3%) | 82 (64.4%) | |
| ≥65 | 406 (48.7%) | 45 (35.4%) | |
| Sex | | | 0.5622 |
| Male | 519 (62.2%) | 83 (65.4%) | |
| Female | 315 (37.8%) | 44 (34.6%) | |
| ASA | | | 0.1391 |
| 1 | 85 (10.2%) | 17 (13.4%) | |
| 2 | 687 (82.4%) | 104 (81.9%) | |
| 3 | 62 (7.4%) | 6 (4.7%) | |
| BMI | | | 0.917 |
| <25 | 592 (71.0%) | 89 (70.1%) | |
| ≥25 | 242 (29.0%) | 38 (29.9%) | |
| Initial CEA | | | 0.0012 |
| <5 | 689 (82.6%) | 89 (70.1%) | |
| ≥5 | 145 (17.4%) | 38 (29.9%) | |
| CCRT | | | 0.4431 |
| Yes | 255 (30.6%) | 34 (26.8%) | |
| No | 579 (69.4%) | 93 (73.2%) | |
| Location of rectal cancer | | | 0.629 |
| Ra | 235 (28.2%) | 39 (30.7%) | |
| Rb | 599 (71.8%) | 88 (69.3%) | |
| Histological type | | | 0.2479 |
| WD | 75 (9.0%) | 13 (10.2%) | |
| MD | 719 (86.2%) | 101 (79.5%) | |
| PD and others | 40 (4.8%) | 13 (10.2%) | |
| pT | | | <0.0001 |
| Tis or T0 | 58 (7.0%) | 2 (1.6%) | |
| T1 | 121 (14.5%) | 3 (2.4%) | |
| T2 | 203 (24.3%) | 17 (13.4%) | |
| T3 | 405 (48.6%) | 95 (74.8%) | |
| T4 | 47 (5.6%) | 10 (7.9%) | |
| LVI | | | <0.0001 |
| Yes | 245 (29.4%) | 69 (54.3%) | |
| No | 589 (70.6%) | 58 (45.7%) | |
| PNI | | | <0.0001 |
| Yes | 100 (12.0%) | 37 (29.1%) | |
| No | 734 (88.0%) | 90 (70.9%) | |
| Distal resection margin | | | 0.0777 |
| Yes | 4 (0.5%) | 3 (2.4%) | |
| No | 830 (99.5%) | 124 (97.6%)0.0 | |
| pN | | | <0.0001 |
| N0 | 573 (68.7%) | 39 (30.7%) | |
| N1 | 185 (22.2%) | 50 (39.4%) | |
| N2 | 76 (9.1%) | 38 (29.9%) | |
| Harvested lymph nodes | | | 0.3459 |
| <12 | 596 (71.5%) | 85 (66.9%) | |

(*Continued*)

**Table 1.** (Continued)

| Feature | No Recurrence (n = 834) | Recurrence (n = 127) | P value |
|---|---|---|---|
| > = 12 | 238 (28.5%) | 42 (33.1%) | |
| pTNM | | | <0.0001 |
| 0 | 55 (6.6%) | 2 (1.6%) | |
| 1 | 264 (31.7%) | 8 (6.3%) | |
| 2 | 254 (30.5%) | 29 (22.8%) | |
| 3 | 261 (31.3%) | 88 (69.3%) | |
| Postoperative chemotherapy | | | <0.0001 |
| Yes | 499 (40.2%) | 107 (15.7%) | |
| No | 335 (59.8%) | 20 (84.3%) | |

ASA, American Society of Anesthesiologists score; BMI, body mass index; CEA, carcinoembryonic antigen; CCRT, concurrent chemoradiotherapy; Ra, peritoneal reflection above; Rb, peritoneal resection below; WD, well-differentiated; MD, moderately differentiated; PD, poorly differentiated; LVI, lymphovascular invasion; pT, pathologic Tumor stage; PNI, perineural invasion; pN, pathologic Node stage; pTNM, pathologic Tumor-Node-Metastasis stage

## Feature importance depending on machine learning methods

Fig 5 shows the respective values of feature importance in accordance with the machine learning models based on permutational importance. The variable with the highest importance was pT, as assessed by SVM, RF, and XGBoost (0.06, 0.12, and 0.13, respectively), whereas pTNM had the highest importance in LR (0.05). In the SVM, pT and sex had the highest values (0.06).

## Discussion

In this study, we analyzed the factors associated with recurrence performed by four machine learning algorithms using 15-years database of consecutive rectal cancer patients who underwent curative surgery. Although SVM showed the best performance (AUC = 0.831), other machine learning methods also had comparable AUC values of more than 0.8. The comparison of the AUC performances among the various machine learning models did not yield a statistically significant difference (p = 0.274). Thus, the focus is primarily on the AUC values. Among the models evaluated, the SVM demonstrated the highest AUC at 0.831, followed by the RF with an AUC of 0.826, LR at 0.811, and XGBoost at 0.804. Based on these results, the SVM can be considered the most effective model for predicting recurrence in this study. In SVM, RF, and XGBoost, pT was the top-ranked feature of importance, whereas pTNM showed the highest feature importance in LR. Their characteristics were similar in terms of pathologic tumor stage. It is strongly suggested that pathologic tumor stage is the most influential predictor of rectal cancer recurrence after curative resection. Tumor stage is a well-known and established prognostic factor for most malignant diseases [19]. Especially in locally advanced rectal cancer, oncologists try to decrease the tumor stage through CCRT because tumor response

**Table 2. The confusion matrix for model performance comparison.**

| Model | Sensitivity (95% CI) | Specificity (95% CI) | Accuracy (95% CI) | AUC (95% CI) |
|---|---|---|---|---|
| LR | 0.769 (0.564–0.910) | 0.683 (0.606–0.752) | 0.694 (0.624–0.758) | 0.811 (0.749–0.864) |
| RF | 0.731 (0.522–0.884) | 0.802 (0.734–0.860 | 0.793 (0.729–0.848) | 0.826 (0.766–0.877) |
| XGBoost | 0.308 (0.143–0.518) | 0.928 (0.878–0.962) | 0.845 (0.786–0.918) | 0.804 (0.741–0.857) |
| SVM | 0.692 (0.482–0.857) | 0.814 (0.747–0.870) | 0.798 (0.734–0.852) | 0.831 (0.770–0.881) |

LR, logistic regression; RF, random forest; XGBoost, extreme gradient boosting; SVM, supportive vector machine; AUC, area under the curve; CI, confidence interval

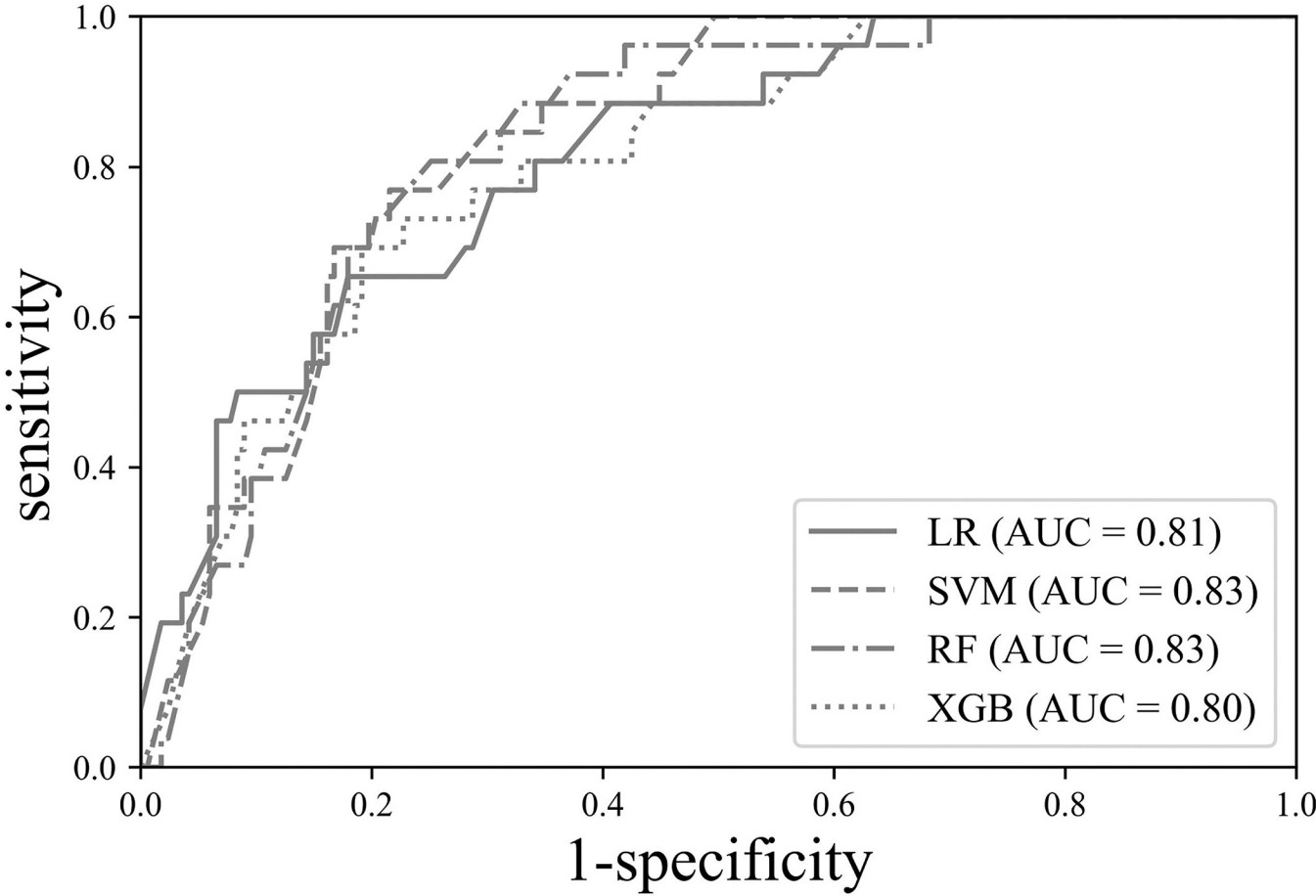

**Fig 4. Receiver operator characteristic (ROC) curve for machine learning models in predicting recurrence after curative rectal cancer surgery.**

with complete response or down-staging provides better oncologic outcomes [20]. In this regard, there are several studies to enhance the efficacy of CCRT with additional preoperative methods [21, 22]. Our findings confirm again that tumor stage is a strongly important factor in the recurrence of rectal cancer.

In all machine learning methods except LR, the first- and second-highest feature importance were pT and sex. According to AJCC 8[th] edition, T3 is defined as 'tumor invades through the muscularis propria into pericolorectal tissues,' and T4a is defined as 'tumor penetrates to the surface of the visceral peritoneum' [12]. Because the lower rectum has no visceral peritoneum, T3 tumors can involve the mesorectal fascia. Therefore, the T stage is a more influential factor in rectal cancer than in colon cancer, which may be reflected in our results. Male sex was another high-ranked risk factor in this study. Previous studies have reported that male sex is a significant predictor for recurrence in colorectal cancer [23–25]. According to Demb et al., male sex had significantly higher odds ratio relative to the female sex for colorectal cancer recurrence, and the odds ratio was higher for rectal cancer (OR = 2.84) compared to the distal colon cancer (OR = 1.84) [25]. This implies that clear surgical resection is more challenging in male patients with rectal cancer because the pelvic cavity in men is narrower and deeper than female patients.

Although CCRT and adjuvant chemotherapy were not high-ranked feature importance in our study, they are conventionally crucial role to improve survival outcomes in rectal cancer. Therefore, we considered distinguishing detailed regimen of perioperative therapies because

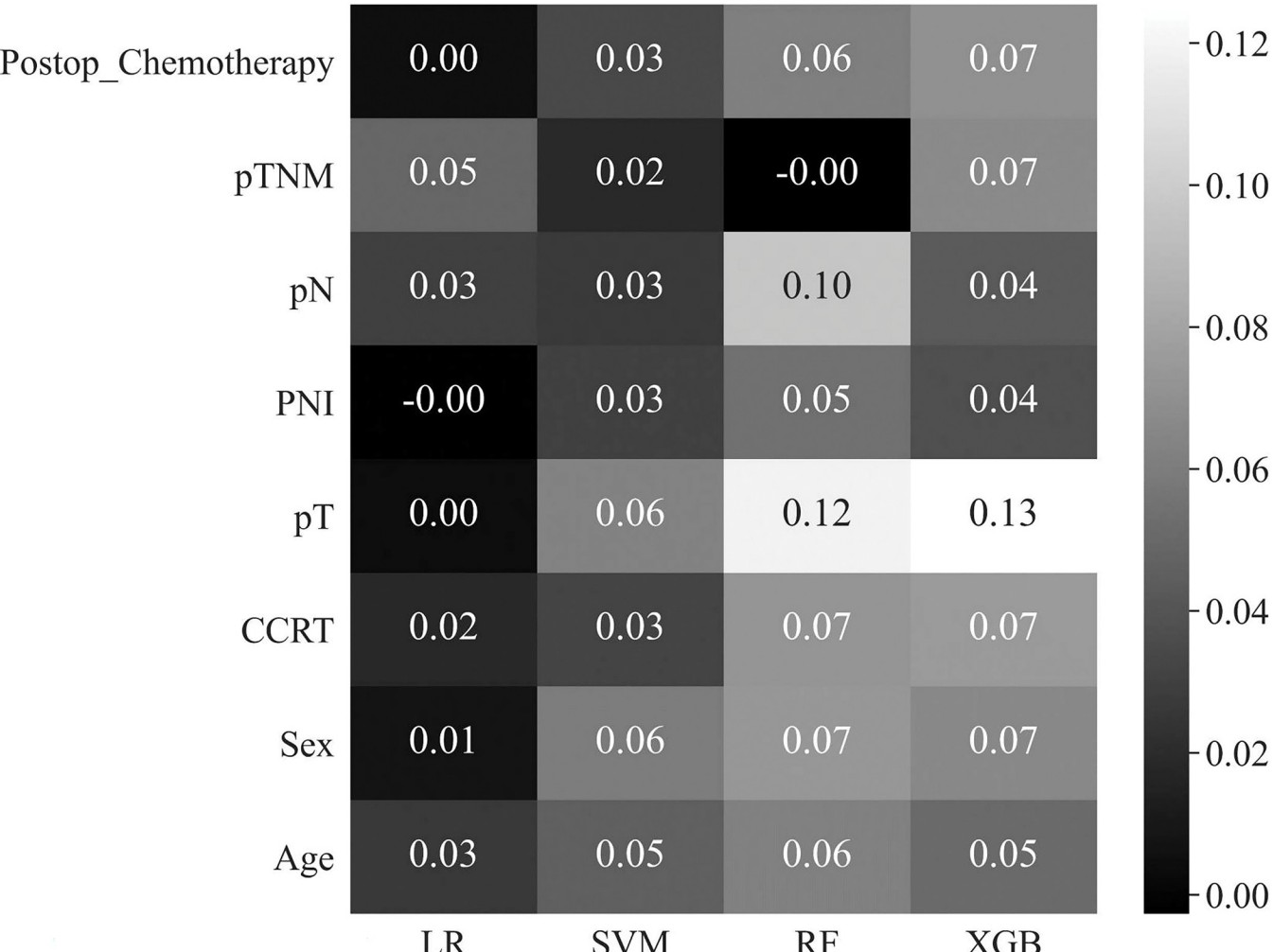

**Fig 5. Feature importance depending on machine learning models.**

anticancer treatment has developed with time. However, consequently, it was no need to classify detailed treatment in this study because we included only non-metastatic rectal cancer patients. Furthermore, every cancer patient except involving clinical trials, received chemotherapy or radiotherapy according to the government guideline, because cancer treatment in South Korea is totally covered by national health insurance. For those reasons, palliative chemotherapeutic agents such as target agents or multikinase inhibitors, and totally neoadjuvant treatment were not considered. From 1980s to now, 5-Fluorouracil/leucovorin have become main regimen of adjuvant chemotherapy for rectal cancer [25–27]. Exceptionally, 5-fluorouracil/leucovorin with oxaliplatin (FOLFOX) regimen has been covered by national health insurance since August 2016, after ADORE study suggested that FOLFOX showed better oncologic outcomes than the conventional 5-fluorouracil/leucovorin for stage II,III rectal cancer patients who received CCRT [28]. In our study, because the patients who received adjuvant FOLFOX chemotherapy after August 2016 were only 4.1% (n = 40/961), it is thought to be a negligible impact to evaluate the predictive factors of recurrence.

All machine learning models performed reliably, with no statistically significant differences in performance (p = 0.274). The SVM demonstrated the highest AUC performance, whereas

the RF may be a better choice when considering sensitivity and specificity. RF achieved the second-best performance with an AUC of 0.826, and the difference between sensitivity and specificity was smaller compared to SVM. SVM exhibited relatively large discrepancies in sensitivity (0.692) and specificity (0.814), indicating the potential presence of bias in training compared to RF. However, owing to the limited size of the test data, it is not possible to definitively conclude that the SVM is more biased. SVM models excel in identifying complex decision boundaries within high-dimensional datasets containing numerous features. This capability stems from their efficient margin maximization between classes, enhancing the model's generalizability and resilience against outliers. However, SVM models face challenges with large datasets due to their time-intensive nature and the complexity involved in selecting optimal kernels and parameters. Conversely, they exhibit high efficiency with smaller datasets. In contrast, tree-based models such as RF and XGBoost, while effective, may encounter overfitting issues in high-dimensional spaces. The distinct attributes of SVM models proved advantageous in our analysis of the colon cancer surgery database. This led to SVM models achieving the highest AUC score, suggesting their superiority in predicting rectal cancer recurrence within the context of this specific database.

This study had several limitations. First, this was a single-center retrospective study, and selection bias could not be excluded. Secondly, the analysis was performed using only a limited number of factors. There were no other clinically significant factors, such as smoking status, tumor regression grade after CCRT, mesorectal fascia involvement, or various molecular biomarker statuses (ras or microsatellite instability). We attempted to analyze as many factors as possible; however, there were many factors with more than 20% missing data. Factors with large proportions of missing data were excluded to improve the quality of the database. Consequently, no data were missing in our study. Third, there was an imbalance in the data ratios between the recurrence and non-recurrence groups. We employed the SMOTETomek technique to address this imbalance; however, it has limitations in fully resolving the underlying problem. The amount of data available for testing in the recurrence group was insufficient for adequate validation. Further research involving cross-validation is required to address these issues. Future studies should focus on collecting additional data from recurrence groups, and the generalization of the model should be addressed through the collection and validation of multicenter data. Finally, we did not distinguish between the p and yp stages (i.e., pathologic findings following preoperative systemic chemotherapy or radiation prior to surgery as a primary treatment) in the pathologic tumor stage. Because the tumor stage could decrease after CCRT, the p-stage could be underestimated in patients treated with CCRT. However, despite of some limitations, we tried to evaluate risk factors for recurrence, focusing on rectal cancer who underwent curative resection, using various machine learning techniques without missing data. Our study has the strength in terms of improving the quality of analysis through multiple machine learning methods, compared to other studies that usually evaluated by single analysis method, LR.

## Conclusions

In this study, we analyzed and compared the importance of risk factors for rectal cancer recurrence using four different machine learning methods. We found that various machine learning methods increased the predictive validity of rectal cancer recurrence. The SVM showed the best AUC value. The most influential factor was pT for all machine learning methods, except for LR. Clinicians should be more alert if patients have a high pT stage during postoperative follow-up.

## Supporting information

**S1 File. Data file.**
(XLSX)

## Acknowledgments

We would like to thank Editage (www.editage.co.kr) for the English language editing.

## Author Contributions

**Conceptualization:** Kwang-Gi Kim, Jeong-Heum Baek.

**Data curation:** Youngbae Jeon, Kug-Hyun Nam, Tae-Sik Hwang.

**Formal analysis:** Youngbae Jeon, Young-Jae Kim, Jisoo Jeon.

**Investigation:** Youngbae Jeon, Young-Jae Kim, Jisoo Jeon.

**Supervision:** Kwang-Gi Kim, Jeong-Heum Baek.

**Writing – original draft:** Youngbae Jeon, Young-Jae Kim.

**Writing – review & editing:** Youngbae Jeon, Young-Jae Kim.

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
