## [Decision Letter · Decision Letter 0]

20 Oct 2023

PONE-D-23-21370Machine learning based prediction of recurrence after curative resection for rectal cancerPLOS ONE

Dear Dr. Baek,

Thank you for submitting your manuscript to PLOS ONE. After careful consideration, we feel that it has merit but does not fully meet PLOS ONE’s publication criteria as it currently stands. Therefore, we invite you to submit a revised version of the manuscript that addresses the points raised during the review process. The comments of two reviewers are listed below. 

We look forward to receiving your revised manuscript.

Kind regards,

Chong-Chi Chiu

Academic Editor

PLOS ONE

Journal Requirements:

Additional Editor Comments:

Please revise your article according to the suggestions of two reviewers.

Reviewers' comments:

Reviewer's Responses to Questions

**Comments to the Author**

1. Is the manuscript technically sound, and do the data support the conclusions?

Reviewer #1: Yes

Reviewer #2: Yes

2. Has the statistical analysis been performed appropriately and rigorously? 

Reviewer #1: I Don't Know

Reviewer #2: I Don't Know

3. Have the authors made all data underlying the findings in their manuscript fully available?

Reviewer #1: Yes

Reviewer #2: Yes

4. Is the manuscript presented in an intelligible fashion and written in standard English?

Reviewer #1: Yes

Reviewer #2: Yes

5. Review Comments to the Author

Reviewer #1: This study aimed to analyze factors related to rectal cancer recurrence after curative resection using different machine learning technique based on the analysis of 961 patients who underwent curative surgery for rectal cancer between 2004 and 2018 at Gil Medical Center. Overall, they found that SVM had best AUC, and the most influential factor across all machine learning methods except LR was found to be pT.

Major comment

1. This finding of this study was based on the data collected between 2004 and 2018. Because anti-cancer treatment had great improvement with time, it is better to use more updated data to establish prediction model.

2. The discussion about the clinical implication may be added.

Minor comment

1. When you introduce an abbreviation in the abstract and text, you should first provide the full term followed by the abbreviation in parentheses.

2. Please add a new figure to reveal the process of patient selection.

3. Please briefly state how to treat patients with rectal cancer in the study site.

3. Please discuss the strength of the present study.

Reviewer #2: This study analyzed factors influencing rectal cancer recurrence after curative surgery using machine learning. It involved 961 patients who underwent surgery, excluding specific cases. Data imbalance was addressed with SMOTETomek. The top eight predictive variables included pT stage, sex, concurrent chemoradiotherapy, pN stage, age, postoperative chemotherapy, pTNM stage, and perineural invasion. Support Vector Machine (SVM) yielded the highest AUC (0.831) for recurrence prediction, with sensitivity, specificity, and accuracy of 0.692, 0.814, and 0.798. The study emphasizes vigilance for patients with high pT stages during postoperative follow-up in rectal cancer cases.

Comments

1. Please spell full name of pT, pN, pTNM for the first time in the abstract section.

2. Please describe how to compare the AUC value among various model and define the best machine learning model using statistical analysis.

3. The underlying conditions and comorbidities of included patients should be added in the table 1.

4. Briefly describe how to manage the patients with rectal cancer in the study site.

5. Add some discussion about why SVM could be the best predicted model.

6. PLOS authors have the option to publish the peer review history of their article (what does this mean?). If published, this will include your full peer review and any attached files.

Reviewer #1: No

Reviewer #2: No

---

## [Author Response · Author response to Decision Letter 0]

4 Dec 2023

● Reviewer #1:

This study aimed to analyze factors related to rectal cancer recurrence after curative resection using different machine learning technique based on the analysis of 961 patients who underwent curative surgery for rectal cancer between 2004 and 2018 at Gil Medical Center. Overall, they found that SVM had best AUC, and the most influential factor across all machine learning methods except LR was found to be pT.

Major comment

1. This finding of this study was based on the data collected between 2004 and 2018. Because anti-cancer treatment had great improvement with time, it is better to use more updated data to establish prediction model.

Answer) 

First of all, we declare that every cancer patient except involving clinical trials received chemotherapy or radiotherapy according to the government guideline, because cancer treatment in South Korea is totally covered by national health insurance. As adjuvant chemotherapeutic regimens for colorectal cancer, 5-FU/leucovorin have covered by national health insurance before 2000s, 5-FU/leucovorin with oxaliplatin (FOLFOX) have covered in the mid-2000s, and capecitabine with oxaliplatin (CAPOX) have covered in the 2010s. Because our study included only non-metastatic rectal cancer, there was no need to consider changes in palliative chemotherapeutic regimen such as target agents or multikinase inhibitors, and there was no significant change in adjuvant chemotherapeutic agents during study periods (CAPOX is only covered in colon cancer). Furthermore, in rectal cancer, totally neoadjuvant treatment (TNT) is not covered by national health insurance. Exceptionally, 5-fluorouracil/leucovorin with oxaliplatin (FOLFOX) regimen has been covered by national health insurance since August 2016, after ADORE study suggested that FOLFOX showed better oncologic outcomes than the conventional 5-fluorouracil/leucovorin for the patients with ypStage Ⅱ,Ⅲ. In current study, because the patients who received FOLFOX chemotherapy after August 2016 were only 4.1% (n=40/961), it is thought to be a negligible impact to evaluate the predictive factors of recurrence. However, your comment is reasonable and very important. Therefore, we added above contents in discussion section (page 17, line 3).

2. The discussion about the clinical implication may be added.

Answer) 

We added more clinical implication including answer of comment number 1 in discussion section.

Minor comment

1. When you introduce an abbreviation in the abstract and text, you should first provide the full term followed by the abbreviation in parentheses.

Answer) 

Including the above, all missing full names of the words have been filled in, and the meaning of the words have been descripted more clearly.

2. Please add a new figure to reveal the process of patient selection.

Answer) 

We added a new figure and figure legends (Figure 1. Flowchart of the patient selection) in method section (page 7, line 4). Consequently, numbering of the following figures have been revised.

3. Please briefly state how to treat patients with rectal cancer in the study site.

Answer) 

We added the process of treating patient with rectal cancer in method section (Patient selection and dataset, page 6, line 13).

3. Please discuss the strength of the present study.

Answer) 

We added the strength of our study in the bottom of discussion section.

● Reviewer #2:

This study analyzed factors influencing rectal cancer recurrence after curative surgery using machine learning. It involved 961 patients who underwent surgery, excluding specific cases. Data imbalance was addressed with SMOTETomek. The top eight predictive variables included pT stage, sex, concurrent chemoradiotherapy, pN stage, age, postoperative chemotherapy, pTNM stage, and perineural invasion. Support Vector Machine (SVM) yielded the highest AUC (0.831) for recurrence prediction, with sensitivity, specificity, and accuracy of 0.692, 0.814, and 0.798. The study emphasizes vigilance for patients with high pT stages during postoperative follow-up in rectal cancer cases.

Comments

1. Please spell full name of pT, pN, pTNM for the first time in the abstract section.

Answer) 

Including the above, all missing full names of the words have been filled in, and the meaning of the words have been descripted more clearly.

2. Please describe how to compare the AUC value among various model and define the best machine learning model using statistical analysis.

Answer) 

In the discussion section, we statistically compared the AUC values among the models and added that SVM is the best recurrence prediction model. (page 15, line 17)

3. The underlying conditions and comorbidities of included patients should be added in the table 1.

Answer) 

We collected the patients’ data from 15 years of electric medical records, and only clearly recorded variables were collected to reduce missing data. Although there was no information of comorbidity in old records, ASA (American Society of Anesthesiologists) classification was definitely recorded in all patients. ASA classification is a method to determine the patients’ systemic condition before surgery under general anesthesia, so we can evaluate the severity of major systemic diseases. Therefore, we did not adopt comorbidity, which had a lot of missing data, as a variable, but adopted ASA classification, which had no missing data. It is believed that ASA classification can generally indicate the patients’ underlying condition and comorbidities.

4. Briefly describe how to manage the patients with rectal cancer in the study site.

Answer) 

We added the process of treating patient with rectal cancer in method section (Patient selection and dataset, page 6, line 13).

5. Add some discussion about why SVM could be the best predicted model.

Answer) 

We added a discussion in the discussion section about why SVM is the best predictive model. (page 18, line 3)

Thank you again for your insightful comments and suggestions. We appreciate you spending your valuable time in revising our paper. 

Sincerely,

---

## [Editor Report · Decision Letter 1]

6 Dec 2023

Machine learning based prediction of recurrence after curative resection for rectal cancer

PONE-D-23-21370R1

Dear Dr. Baek,

We’re pleased to inform you that your manuscript has been judged scientifically suitable for publication and will be formally accepted for publication once it meets all outstanding technical requirements.

Kind regards,

Chong-Chi Chiu

Academic Editor

PLOS ONE

---

## [Editor Report · Acceptance letter]

8 Dec 2023

PONE-D-23-21370R1 

Machine learning based prediction of recurrence after curative resection for rectal cancer 

Dear Dr. Baek:

I'm pleased to inform you that your manuscript has been deemed suitable for publication in PLOS ONE. Congratulations! Your manuscript is now with our production department. 

Kind regards, 

on behalf of

Professor Chong-Chi Chiu 

Academic Editor

PLOS ONE